# Exploiting Latent Information in Relational Databases via Word Embedding

**Rajesh Bordawekar**
IBM T. J. Watson Research Center
Yorktown Heights, NY 10598, USA
bordaw@us.ibm.com

**Oded Shmueli** [*]
Computer Science Department
Technion, Haifa 32000, Israel
oshmu@cs.technion.ac.il

**Bortik Bandyopadhyay** [†]
Computer Science Department,
Ohio State University, Columbus, OH 43210, USA
bandyopadhyay.14@osu.edu

## Abstract

We propose Cognitive Databases, an approach for transparently enabling Artificial Intelligence (AI) capabilities in relational databases. A novel aspect of our design is to first view the structured data source as meaningful unstructured text, and then use the text to build a word embedding model. This model captures the hidden inter-/intra-column relationships between database tokens of different types such as numeric values, SQL Dates, and even images. For each database token, the model includes a vector that encodes contextual semantic relationships. We seamlessly integrate the word embedding model into existing SQL query infrastructure and use it to enable a new class of SQL-based analytics queries called cognitive intelligence (CI) queries. CI queries use the model vectors to enable complex queries such as semantic similarity/dissimilarity, inductive reasoning queries such as analogies and semantic clustering, predictive queries using entities not present in a database, and, more generally, using knowledge from external sources.

## 1 Introduction

Traditionally, relational databases have been used to analyze enterprise datasets that comprise mostly of well-qualified typed entities (e.g., character(n), decimal, float, or timestamp). However, over the years, relational databases have been increasingly used to store and process free-formed unstructured text data (e.g., customer reviews). It is intuitively clear that databases with such unstructured text entities have a significant amount of latent semantic information. However, columns that contain different types of data, e.g., strings, numerical values, images, dates, etc., possess significant latent information in the form of inter- and intra-column relationships. The usual way to utilize this information is using SQL and extensions, such as text extensions, or User Defined Functions (UDFs) to handle exotic data types. However, these extensions are rather limited in their *smarts*. Specifically, SQL queries rely on *value-based* analytics to detect patterns. In addition, the relational data model neglects many inter- or intra-column relationships. Thus, the traditional SQL queries lack a *holistic* view of the underlying relations, and thus are unable to extract and exploit semantic relationships that are *collectively* generated by tokens in a database relation.

This paper discusses Cognitive Database (Bordawekar et al., 2017; Bordawekar & Shmueli, 2017), a novel relational database system, which uses word embedding techniques (Mikolov et al., 2013b;a; Levy & Goldberg, 2014) to extract latent knowledge from a database table. The generated word-embedding model captures inter- and intra-column semantic relationships between database tokens of different types. For each database token, the model includes a vector that encodes contextual semantic relationships. The cognitive database seamlessly integrates the model into the existing

---

[*] Work done while the author was visiting IBM Research.
[†] Work done while the author was visiting IBM Research.

SQL query processing infrastructure and uses it to enable a new class of SQL-based analytics queries called Cognitive Intelligence (CI) queries. CI queries use the model vectors to enable complex semantic queries over relational data such as semantic similarity or dissimilarity, inductive reasoning queries such as analogies or semantic clustering, and predictive queries using entities not present in a database.

## 2 Cognitive Database Design

In the database context, vectors may be produced by either learning on text transformed and extracted from the database itself and/or using external text sources, such as wikipedia. Training a word-embedding model from a relational database requires two stages. The first stage, *textification*, takes a relational table with different SQL types as input and returns an unstructured but meaningful text corpus consisting of a set of sentences. This transformation allows us to generate a multi-modal embedding model with uniform semantic representation of different SQL types. In addition to text tokens, our current implementation supports numeric values and images (we assume that the database being queried contains a `VARCHAR` column storing links to the images). We use different strategies for converting a non-text relational data to text: e.g., values in a numeric column are first clustered using a standard clustering approch (e.g., K-Means), and then replaced by a text token that represents the corresponding cluster. For images, one approach classifies images into classes using a pre-trained model and then represents each image by a string token that represents its class. Alternatively, one can first extract text features from an image using off-the-shelf image services, such as IBM Watson Visual Recognition Service (IBM Watson), and then use the extracted features to train the embedding model.

We use an unsupervised training approach based on the Word2Vec (W2V) (Mikolov, 2013) implementation to build the word embedding model from the generated text corpus. The text corpus is organized as a set of *English-like* sentences, separated by stop words (e.g., newline). Each sentence correspond to a row in the relational view and used as a neighborhood context during the training of the word embedding model. Hence, *the inferred semantic meaning of the relational entities reflect the collective relationships defined by the associated relational view (generated by relational operations such SELECT, PROJECT, and JOIN.)*

Our training implementation builds on the classical W2V implementation, but it varies from the classical approach in a number of important aspects: (1) A sentence generated from a relational row is generally not in any natural language such as English. Therefore, W2Vs assumption that the influence of any word on a nearby word decreases as the word distances increases, is not applicable. In our implementation, every token in the training set has the same influence on the nearby tokens in the context. (2) Another consequence is that unlike an English sentence, the last word is equally related to the first word as to its other neighbors. To enable such relationships for the last word, the first word can be viewed as its immediate neighbor). (3) For relational data, we provide special consideration to primary keys, which can be unique. First, the classical W2V discards less frequent words from computations. In our implementation, every token, irrespective of its frequency, is assigned a vector. Second, irrespective of the distance, a primary key is considered a neighbor of every other word in a sentence and included in the neighbor-hood window for each word. Also, the neighborhood extends via foreign key occurrences of a key value to the row in which that value is key. (4) Finally, our implementation is designed to enable incremental training, i.e. the training system takes as input a pre-trained model and a new set of generated sentences, and returns an updated model. This capability is critical as a database can be updated regularly and one can not rebuild the model from scratch every time. The use of pre-trained models is an example of transfer learning, where a model trained on an external knowledge base can be used either for querying purposes or as a basis of a new model.

## 3 Cognitive Intelligence Queries

The cognitive relational database has been designed as an extension to the underlying relational database, and thus supports all existing relational features. The cognitive relational database supports a new class of business intelligence (BI) queries called Cognitive Intelligence (CI) queries. The CI queries extract information from a relational database based, in part, on the contextual seman-

tic relationships among database entities, encoded as meaning vectors. At runtime, the SQL query execution engine uses various user-defined functions (UDFs) that fetch the trained vectors from the system table as needed and answer CI queries. The CI queries take relations as input and return a relation as output. CI queries augment the capabilities of the traditional relational BI queries and can be used in conjuction with existing SQL operators.

Our current implementation is built on the Apache Spark 2.2.0 infrastructure. It supports four types of CI SQL queries: similarity queries, inductive reasoning, prediction, and cognitive OLAP. These queries can be executed over databases with multiple datatypes: we currently support text, numeric, and image data. The similarity queries compare two relational variables based on similarity or dissimilarity between the input variables. Each relational variable can be either set or sequence of tokens. In case of sequences, computation of the nal similarity value takes the ordering of tokens into account. The similarity value is then used to classify and group related data. The inductive reasoning queries exploit latent semantic information in the database to reason from part to whole, or from particular to general (Sternberg & Gardner, 1979; Rumelhart & Abrahamson, 1973). We support different types of inductive reasoning queries: analogies, semantic clustering, analogy sequences, clustered analogies, and odd-man-out. Given an item from an external data corpus (which is not present in a database), the predictive CI query can identify items from the database that are similar or dissimilar to the external item by using the externally trained model. Finally, cognitive OLAP allows SQL aggregation functions such as MAX(), MIN() or AVG() over a set that is identified by contextual similarity computations on the relational variables.

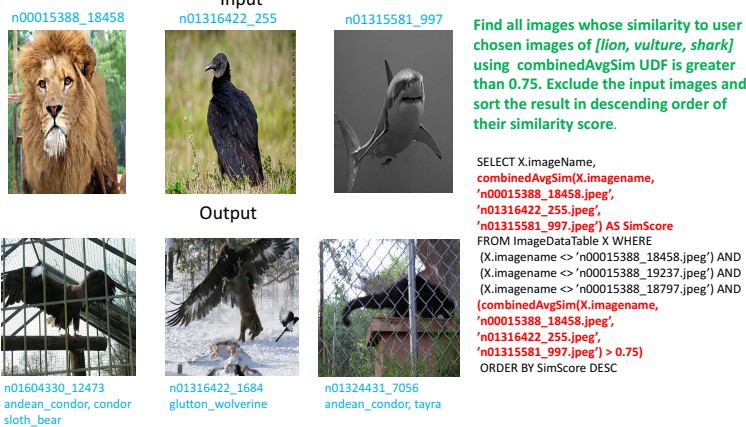

Figure 1: Inductive reasoning CI query for semantic clustering of images

To demonstrate the capabilities of the Cognitive Database, consider a semantic clustering CI query on a relational *multi-modal* database (Figure 1 ): the original database lists national parks with string tokens representing image file names, e.g., n00015388_18458.jpeg. We first create a training table using text features extracted from the images using the Watson VRS system. The training table is then used to build a multi-modal word embedding model that captures relationalships between text and image features. This model is then used to answer CI queries that use both text and image variables. For example, the goal of query shown in Figure 1 is to identify all images that are similar to every image in the set of user chosen images. Such images share one or more features with the input set of images. For this query, we select images of a lion, a vulture, and a shark as the input set and use the combinedAvgSim() UDF to identify images that are similar to all these three images. Although the input images display animals from three different classes, they share one common feature: all three animals are carnivorous. The UDF computes the average vector from the three input images and then selects those images whose vectors are similar to the computed average vector with similarity score higher than 0.75. Figure 1 shows the top three image results: andean condor, glutton wolverine, and tyra. Although these animals are from different classes, they all are carnivores, a feature that is shared with the animals from the input set.

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
