# OpenReview forum: "EXPLOITING LATENT INFORMATION IN RELATIONAL DATABASES VIA WORD EMBEDDING"
_ICLR.cc/2018/Workshop — Reject_

### Official Review · AnonReviewer1 · 2018-03-09
**Interesting direction, but too preliminary**

**Rating:** 5
**Confidence:** 4

**Review:**

This paper presents an approach for learning embeddings of database records, to allow new types of queries on the database such as similarity queries.

I've seen word2vec approaches used on tabular data before (like tables from the Web), where the context of a cell is the other cells in its row and column.  However, the four adaptations of w2v to this domain listed in the final paragraph of Section 2 seem novel and appropriate, and the application to the database domain -- e.g. multiple relational views from the same database -- is more novel.

The limitation of this paper is that the ideas seem underdeveloped, even for the workshop stage.  I think the first two paragraphs of the paper could be improved by discussing one compelling, concrete example of the kind of query we can answer with this new approach that is difficult or impossible to answer with other techniques, like mining association rules from the database or by propositionalizing the relational structure and applying standard machine learning.  The one example query we're shown in the paper (finding what all three images have in common, that they are pictures of carnivores) seems like something that could be achieved with a non-embedding approach that mined associations in the relational data, provided we had a database saying which animal was in each image and also characteristics of the animals.

I also feel that the naming (referring to the new queries as "cognitive intelligence" queries) oversells the approach a bit.

Minor:
each sentence correspond -> each sentence corresponds
what about handling deletes from a database?
computation of the nal similarity -- I wasn't sure what "nal" meant

---

### Official Review · AnonReviewer3 · 2018-03-10
**Database queries with word embeddings**

**Rating:** 4
**Confidence:** 4

**Review:**

This paper describes an extension of SQL queries using word embeddings. By apply word2vec to relational database entries (i.e. tuples) , database entries can be matched when their embedding vectors are similar.

The research direction is interesting and may be useful in practice. However, it is not innovative enough to be presented at a workshop targeting  representation learning.

---

### Official Review · AnonReviewer2 · 2018-03-11
**Limited novelty and lacks experimental details**

**Rating:** 4
**Confidence:** 4

**Review:**

This paper applies word2vec model to relational databases and calls the resulting system Cognitive Databases. While the CI queries part is interesting, the abstract is shallow in terms of details. Moreover, apart from a few qualitative examples, there is no quantitative comparison to demonstrate utility of the proposed approaches. In the last paragraph of Sec 2, the authors try to distinguish their approach from W2V. However, I don't find those distinctions to be significant. For example, importance of all the words in the sentence may be achieved by increasing context window size in W2V. Similarly, incremental training is entirely possible (and a routine thing to do) while fine tuning W2V to a new corpus.

Overall, I feel the use of embeddings in relational databases is interesting. However, the proposed method lacks novelty and experimental details and is not ready as is.

---

### Decision · Program_Chairs · 2018-03-20
**ICLR 2018 Workshop Acceptance Decision**

**Decision:**

Reject

**Comment:**

Based on the reviews, this paper has not been accepted for presentation at the ICLR workshop. However, the conversation and updates can continue to appear here on OpenReview.